# Soluble Polymer Microneedles Loaded with Interferon Alpha 1b for Treatment of Hyperplastic Scar

**DOI:** 10.3390/polym15122621

**Published:** 2023-06-08

**Authors:** Baorui Wang, Suohui Zhang, Aguo Cheng, Juan Yan, Yunhua Gao

**Affiliations:** 1Key Laboratory of Photochemical Conversion and Optoelectronic Materials, Technical Institute of Physics and Chemistry of Chinese Academy of Sciences, Beijing 100190, China; wangbaorui18@mails.ucas.ac.cn (B.W.); chengaguo19@mails.ucas.ac.cn (A.C.); 2School of Future Technology, University of Chinese Academy of Sciences, Beijing 100049, China; 3Beijing CAS Microneedle Technology Ltd., Beijing 102609, China; yj1010364294@163.com; 4College of Life Sciences, Changchun Normal University, Changchun 130032, China

**Keywords:** hyperplastic scar, interferon alpha 1b, transdermal drug delivery, polymeric microneedles, carboxymethyl cellulose

## Abstract

To achieve the painless administration of interferon alpha 1b (rhIFNα-1b), a double-layered soluble polymer microneedle (MN) patch loaded with rhIFNα-1b was used to deliver rhIFNα-1b transdermally. The solution containing rhIFNα-1b was concentrated in the MN tips under negative pressure. The MNs punctured the skin and delivered rhIFNα-1b to the epidermis and dermis. The MN tips implanted in the skin dissolved within 30 min and gradually released rhIFNα-1b. The rhIFNα-1b had a significant inhibitory effect on the abnormal proliferation of fibroblasts and excessive deposition of collagen fibers in the scar tissue. The color and thickness of the scar tissue treated using the MN patches loaded with rhIFNα-1b were effectively reduced. The relative expressions of type I collagen (Collagen I), type III collagen (Collagen III), transforming growth factor beta 1 (TGF-β1), and α-smooth muscle actin (α-SMA) were significantly downregulated in scar tissues. In summary, the MN patch loaded with rhIFNα-1b provided an effective method for the transdermal delivery of rhIFNα-1b.

## 1. Introduction

A hyperplastic scar is a fibrotic disease of the skin caused by the abnormal proliferation of fibroblasts and excessive deposition of collagen [1,2]. Fibroblasts proliferate abnormally during wound healing and secrete excessive collagen, causing the formation of scars [3,4]. Scars from trauma, surgery or burn cause serious physical and psychological damage to patients [5]. Currently, scar treatment methods can be divided into surgical and conservative treatments [6,7]. Common surgical treatments include scar excision, Z-plasty, W-plasty, and skin graft [8]. The complexity of surgery and secondary injury limit the use of surgical treatments. Common conservative treatments include corticosteroids, compression, laser, and irradiation [9,10]. Conservative treatments with higher patient compliance are considered superior to surgical treatments [11]. However, the recurrence rate of conservative treatment is higher [12]. The side effects of corticosteroids and irradiation on patients are also gaining attention [13,14].

Interferons are a class of secreted glycoproteins with antiviral and antiproliferative properties [15]. Interferons induce apoptosis to reduce fibroblast proliferation, and increase collagenase activity to downregulate collagen production [16,17]. Different interferon subtypes have different therapeutic effects and recurrence rates for hyperplastic scars [18,19]. According to related studies, the recurrence rate of subcutaneous rhIFNα-1b for scars was statistically significantly lower than that of surgical treatment [20]. The rhIFNα-1b was also an effective adjuvant therapy to surgical treatment, with a lower recurrence rate than surgical treatment alone [21]. Although rhIFNα-1b is effective in treating scars, it requires frequent injections during treatment, which not only irritates scar tissue but also decreases patient compliance [22,23]. Therefore, it is important to develop a painless and convenient formulation of rhIFNα-1b.

Oral and transdermal drug delivery are the main non-invasive administration types [24,25]. The rhIFNα-1b can be catabolized by gastric acid, so it cannot be directly administered orally. The stratum corneum is the natural barrier of the skin and has a barrier effect on transdermal drug delivery [26]. Conventional patches or ointments are also unable to deliver rhIFNα-1b transdermally. The microneedle (MN) patch is a painless, minimally invasive, self-administered transdermal drug delivery preparation consisting of micron-sized needles arranged on a patch [27,28]. The MN tips can puncture the stratum corneum of the skin and form an array of micropores for drug delivery in the skin [29,30]. The MN patch is a preparation that physically facilitates transdermal drug delivery, significantly increases the efficiency of transdermal drug delivery, greatly expands the range of drugs that can be delivered transdermally, and it has a great potential to replace injections [31,32].

According to the differences in drug loading and delivery, MNs can be classified as solid MNs, coated MNs, hollow MNs, dissolving MNs, and hydrogel-forming MNs [33,34]. Among them, the development of dissolving MNs is an important breakthrough in the field of MN drug delivery. Biodegradable or dissolvable biocompatible materials have been used as substrate materials for dissolving MNs [35,36]. Dissolving MNs are less costly and do not generate biohazardous waste during manufacturing, storage, and transportation [37,38]. Therefore, dissolving MNs are considered to be the most promising MNs for industrialization [39]. Dissolving MNs have been extensively studied for the transdermal delivery of a wide range of drugs, including small molecules, proteins, nucleic acids, vaccines, and nano-sized particles [40,41].

Here, our group developed a double-layered soluble polymer MN patch loaded with rhIFNα-1b. The first layer of the solution contained rhIFNα-1b and carboxymethyl cellulose (CMC, 8000 cp). The CMC (8000 cp) has a slow dissolution rate in the skin and has the ability to release rhIFNα-1b slowly. During MN fabrication, the first layer of solution was concentrated in the MN tips under negative pressure. The drug loaded in the MN tips can be implanted directly into the skin. The drug loaded in the MN tips has a higher drug delivery efficiency than the drug loaded in the MN backing. The second layer of solution contained a high concentration of CMC (36 cp). After the MN tips punctured the skin, the MN backing dissolved due to contact with the subcutaneous tissue. The MN tips were separated from the MN backing, leaving the MN tips loaded with rhIFNα-1b in the skin. The faster dissolution rate of CMC (36 cp) in the skin allowed for the rapid separation of the MN tips from the MN backing. The shape, mechanical strength, skin puncture, and puncture depth of the MNs were characterized in vitro. The dissolution rate, delivery efficiency, drug release rate, analysis of inflammation, deposition of collagen, type of collagen, and gene expression were characterized in vivo. The MN patch loaded with rhIFNα-1b is an effective transdermal delivery method for rhIFNα-1b, offering a new option for the treatment of hyperplastic scars.

## 2. Materials and Methods

### 2.1. Materials and Animals

CMC (36 cp and 8000 cp) were purchased from Shanhe Pharmaceutical Excipients Co., Ltd. (Anhui, China). The rhIFNα-1b concentrate was provided by Tri-Prime Gene Pharmaceutical (Beijing, China). Triamcinolone was purchased from Heowns Biochemical Technology Ltd. (Tianjin, China). Fluorescein isothiocyanate-labelled bovine serum albumin (BSA-FITC) was purchased from Solarbio (Beijing, China). Fresh porcine ear skin was purchased from a local slaughterhouse. Fresh porcine ear skin was purchased from Kaikai Technology Trading Co., Ltd. (Shanghai, China). ICR mice were obtained from Charles River (Beijing, China). Male New Zealand white rabbits were obtained from Jinmuyang Laboratory Animal Breeding Co., Ltd. (Beijing, China).

### 2.2. MN Fabrication Process

To enrich rhIFNα-1b at the MN tip, a two-step mold casting method was used to fabricate a soluble polymer MN patch. An aqueous solution containing 3.75 g/L rhIFNα-1b and 1.6% (*w*/*w*) CMC (8000 cp) was used to fabricate the first layer. A total of 40 μL of the first layer solution was dropped onto a polydimethylsiloxane (PDMS) mold. The shape of the microneedle patch was designed by Beijing CAS Microneedle Technology Ltd. A total of 117 MNs were arranged on a 6 mm × 6 mm patch. The distance between the MN tips was 500 μm. The MNs were cones of 700 μm in height. Subsequently, the mold was air dried at room temperature for 1 h, maintaining a pressure of −0.1 MPa under the molds. During the drying process, the first layer solution entered the pinholes of the mold under negative pressure. The second layer solution was an aqueous solution containing 20% (*w*/*w*) CMC (36 cp). A total of 40 μL of the second layer solution was dropped onto the mold. A pressure of −0.1 MPa was applied under the mold and maintained for 10 min. The mold with the solution was air dried at room temperature for 1 h and transferred to a desiccator containing silica gel desiccant for 24 h to dry the MN patch loaded with rhIFNα-1b completely.

### 2.3. Characterization of Shape and Skin Puncture

The shape of the MN patch was characterized using an optical microscope (BX51, Olympus, Tokyo, Japan) and stereomicroscope (SMP1000, Nikon, Tokyo, Japan). A force-displacement test machine (AL-5K, SA Precision Instruments, Shanghai, China) was used to detect the displacement of MNs under the force of 0–50 mN/needle. Isolated porcine ear skin was used to evaluate the mechanical strength of MNs. The porcine ear skin stored at −20 °C was thawed in phosphate buffer (pH = 7.4). A MN patch was punctured through the porcine ear skin with a force of 5.85 N, assisted by the force-displacement test machine. The surface of the porcine ear skin was stained with 1% trypan blue dye for 30 min to show the array of pinholes in the skin that were punctured by the MNs [42].

The skin punctured by the MN patch was fixed with 4% paraformaldehyde to prepare longitudinal paraffin sections. Hematoxylin–eosin (HE) dye was used to stain the sections of skin tissue. An optical microscope was used to observe the state of the skin punctured by the MNs [41].

During the fabrication process, rhIFNα-1b was replaced with BSA-FITC to fabricate MNs loaded with 150 μg BSA-FITC. The porcine ear skin was punctured with a MN patch loaded with 150 μg of BSA-FITC. A confocal laser scanning microscope (A1RMP, Nikon, Tokyo, Japan) was scanned at a frequency of 5 μm each time to obtain 3D reconstruction images of the skin punctured by the MNs. The wavelength of excitation light was 488 nm [25].

### 2.4. In Vivo Dissolution

Eighteen male ICR mice (6 weeks old, 23.5 ± 4.6 g) were randomly assigned to six groups to evaluate the rate of MN dissolution in the skin. All animals were kept under 12 h light–dark cycles with free access to food and water. The MN patch loaded with 150 μg rhIFNα-1b was applied to the abdominal skin of male ICR mice. The MNs were implanted into the abdominal skin of male ICR mice and maintained for 0, 2, 5, 10, 20, and 30 min, respectively. The dissolution of MNs in the skin was observed by an optical microscope.

### 2.5. In Vivo Release

Six male ICR mice (6 weeks old, 25.1 ± 5.9 g) were used to evaluate drug release from the MN patch. The MN patch loaded with 150 μg BSA-FITC was applied to the abdominal skin of male ICR mice to observe the drug release from the MN patch. The mice were anesthetized with isoflurane. The whole-body fluorescence imaging of mice was performed via an in vivo FX Pro imaging system (Eastman Kodak, New York, NY, USA) [43].

### 2.6. Scar Modeling and Treatment

Nine male New Zealand White rabbits were randomly assigned to three groups. New Zealand White rabbits were anesthetized with chloral hydrate. The ventral hair of the left ear was removed and disinfected with ethanol (75%, *v*/*v*). The full-thickness skin and perichondrium were separated by a circular cutter with diameter of 10 mm. Each rabbit created four wounds in the left ear. The distance between the wounds was about 1.5 cm. The wounds were hemostatic by gauze compression, and erythromycin ointment was applied to the wounds to prevent infection. After 28 days, hyperplastic scars that were significantly higher than normal skin tissue formed on the skin. Two groups of rabbits received the MN patch loaded with interferon and triamcinolone injection (40 g/L, 0.5 mL), respectively. The MN patch was attached for 8 h. The rabbits received treatment once a week. One group of rabbits was the control group that did not receive any treatment.

### 2.7. Pathological Analysis

The scar tissue of New Zealand White rabbit ears was collected and washed with saline. It was fixed in 4% paraformaldehyde and prepared into 2 μm thick paraffin sections along the longitudinal direction of the skin. HE staining was used for the inflammatory analysis of scar tissue. Masson trichrome staining was used to analyze the morphology of fibroblasts and the collagen deposition. Sirius Red staining was used to discriminate between Collagen I and Collagen III. HE and Masson’s trichrome stained sections were observed under an optical microscope. Sirius Red stained sections were observed under a polarizing microscope (BX51, Olympus, Tokyo, Japan).

### 2.8. Gene Expression Analysis

The scar tissue was stored at −80 °C. In total, 20 mg of scar tissue was ground in liquid nitrogen. Total RNA was extracted from the scar tissue using the SV Total RNA Isolation System (Bio-Rad CFX96 system, Bio-Rad, Hercules, CA, USA). The purity of RNA was evaluated by the ratio of 260 nm/280 nm absorption. The PrimeScript^TM^ RT Reagent Kit (TaKaRa, Beijing, China) was used to synthesize cDNA by reverse transcription, and 2 μg of total RNA was reverse transcribed by the Oligo (dT) 15 Primer (TaKaRa, Beijing, China). The cDNAs from the internal reference gene and genes of interest were PCR amplified in an Agilent Mx3000P Real-Time PCR System using TB Green^TM^ Premix Ex Taq^TM^ II (TaKaRa, Beijing, China). Primer sequences were designed using Primer Premier 6 (Applied Biosystems) and all primers were synthesized by Sangon Biotech (Shanghai, China). The information about the primers is shown in Table 1. Glyceraldehyde 3-phosphate dehydrogenase (GAPDH) was used as the internal reference. The relative expression levels were calculated using the 2^−∆∆CT^ method [23].

### 2.9. Data Analysis

All data are presented as mean  ±  standard deviation. One-way analysis of variance (ANOVA) was used to determine the differences between groups. 

## 3. Results and Discussion

### 3.1. MN Shape and Mechanical Strength

The MN patch loaded with rhIFNα-1b has homogeneous MN tips (Figure 1A). The distance between the MN tips was 491.46 ± 13.72 μm (Figure 1B). The MNs were 680.31 ± 31.87 μm in height (Figure 1C). The mechanical strength of the MN decreased with the increasing amount of rhIFNα-1b in the MN within the range of 0–150 mg per patch (Figure 1D). There was no inflection point in the force–displacement curve, indicating that the MN did not break under axial forces of 0–50 mN/needle. Under a 50 mN/needle force, the displacement of MN loaded with 150 mg/patch of rhIFNα-1b was 40.67 μm more than MN without rhIFNα-1b (Figure 1E). The addition of rhIFNα-1b weakened the mechanical strength of the MN. The displacement of an MN patch loaded with 150 mg rhIFNα-1b under different forces was shown. Under a force of 50 mN/needle, the displacement of the MN was 108.24 mm, and the compression ratio was only 15.46% (Figure 1F). Referring to previous studies, we inferred that the MN patch loaded with rhIFNα-1b had sufficient mechanical strength to puncture the skin. The porcine ear skin punctured by a MN patch loaded with 150 mg rhIFNα-1b showed a clear array of pinholes (Figure 1G). Longitudinal sections of the skin indicated that the MN punctured the stratum corneum of the skin to reach the epidermis and dermis (Figure 1H).

### 3.2. In Vivo Dissolution of MNs

The MN tip implanted in the skin gradually dissolved through the penetration of the microchannel and the hydration of the skin. The MN tip has a larger specific surface area in contact with the subcutaneous tissue, resulting in a faster dissolution rate (Figure 2A). Within 30 min, the height of the MN tip implanted in the skin was reduced by 88.70% (Figure 2B). The substrate material dissolved and released the rhIFNα-1b from the MN tips. The release rate of rhIFNα-1b from the MN patch was influenced by the water solubility of the substrate material. RhIFNα-1b released from the MN patch was absorbed by the capillaries and entered the circulatory system [44,45]. The loading of rhIFNα-1b in the MN patch was adjusted by changing the concentration of rhIFNα-1b in the first layer of solution. The transdermal delivery efficiency of the MN patch loaded with 20–150 mg rhIFNα-1b ranged from 73.60% to 83.72% (*p* > 0.05, Figure 2C). The MN fabrication method was controlled in terms of drug loading and had a stable delivery efficiency.

### 3.3. Depth of Skin Punctured

The depth of skin puncture by the MNs depended on the shape of the MN, the substrate material, and the drug. The porcine ear skin was punctured by a MN patch loaded with BSA-FITC. A confocal laser scanning microscope performed fluorescence imaging of the skin punctured by the MNs. The BSA-FITC loaded in the MN was delivered to 0–300 μm subcutaneously (Figure 3A). The drug in the MN was mainly delivered into the skin along the micropores (Figure 3B).

### 3.4. In Vivo Release of BSA-FITC from the MNs

The MN loaded with BSA-FITC punctured the abdominal skin of ICR mice and delivered BSA-FITC into the skin (Figure 4A,B). The whole-body fluorescence imaging of mice allowed for the direct observation of drug release from the MN patch (Figure 4C). The release rate of the drug in the MN patch was faster during 0–2 days. The sum fluorescence intensity at the MN patch application site decreased by 75.07% and the fluorescence area decreased by 73.08% (Figure 4D,E). The release rate of the drug in the MN patch was slow during 2–7 days. The sum fluorescence intensity at the MN patch application site decreased by 22.43% and the fluorescence area decreased by 25.38%. The reduction in sum fluorescence intensity and fluorescence area indicated that the drug in the MN was released mainly within 0–2 days. In vivo imaging is a semi-quantitative measurement to evaluate the release rate of model drugs in vivo. Since the molecular weight of rhIFNα-1b is smaller than that of BSA-FITC, the release rate of rhIFNα-1b will be faster than that of BSA-FITC.

### 3.5. Scar Status

The status of the rabbit ear scar tissue during treatment was demonstrated (Figure 5). The wound of the rabbit ear healed completely within 28 days, and a light red hyperplastic scar that was significantly thicker than the normal skin tissue was formed on the skin. A triamcinolone injection was used as a reference preparation. After 6 weeks of treatment with the reference preparation or the MN patch loaded with rhIFNα-1b, the scar tissue became significantly lighter in color and approached the color of normal skin. The thickness of the scar tissue was also reduced compared to the pre-treatment. The proliferation of fibroblasts was inhibited by triamcinolone and the rate of degradation was accelerated, resulting in the flattening, and softening of the scar tissue. Triamcinolone is a glucocorticoid. Frequent local injections of triamcinolone may cause lagging skin blanching and mild muscle atrophy at the injection site, and, in a few cases, itching and redness at the injection site. Triamcinolone injection and the MN patch loaded with rhIFNα-1b were both effective in reducing the color and thickness of the scar tissue.

### 3.6. Analysis of Inflammation

The abnormal proliferation of fibroblasts played an important role in the formation of the scars. The aggregation and interaction of inflammatory cells with fibroblasts led to a chronic inflammatory response. HE-stained longitudinal sections of the scar tissue were used for inflammatory analysis. Normal skin tissue had a thin dermis with collagen fibers aligned parallel to each other (Figure 6A). The number of fibroblasts was significantly increased and the collagen fibers were disorganized in the untreated scar tissue (Figure 6B). Inflammatory cell infiltration and edema occurred in the local area. Compared to the untreated scar tissue, the scar tissue treated with the MN patch loaded with rhIFNα-1b had a thinner dermis, a parallel alignment of collagen fibers, and displayed a significant reduction in inflammation and edema (Figure 6C). The triamcinolone injection also had significant repair effects on collagen fibers and inflammation in the scar tissue (Figure 6D). The MN patch loaded with rhIFNα-1b, and the triamcinolone injection both had significant inhibitory effects on the abnormal proliferation of fibroblasts.

### 3.7. Deposition of Collagen

Another important indicator of scar formation was excessive collagen deposition. Collagen is one of the main components of the extracellular matrix. Masson’s trichrome staining was used for the semi-quantitative analysis of muscle and collagen fibers in the scar tissue. The muscle fibers were stained red and the collagen fibers were stained blue. The collagen fiber bundles of normal skin tissue were small and arranged in an orderly and neat manner (Figure 7A). Compared to normal skin tissue, the untreated scar tissue had a significantly increased collagen volume fraction (CVF) and disorganized collagen fibrils (Figure 7B). The CVF was significantly reduced and collagen fibrils were aligned and thickened in the scar tissue treated with the MN patch loaded with rhIFNα-1b compared to the untreated scar tissue (Figure 7C). There was no significant difference in the CVF of the scar tissue treated with triamcinolone injection and the MN patch loaded with rhIFNα-1b (Figure 7D). The MN patch loaded with rhIFNα-1b and triamcinolone injection both had significant inhibitory effects on the excessive deposition of collagen fibers (Figure 8).

### 3.8. Type of Collagen

Sirius Red staining is one of the effective methods of distinguishing collagen types. Different types of collagen fibers have different colors and morphologies under a polarizing microscope. Sirius Red stained Collagen I yellow and red, and Collagen III green. The collagen fibers in normal skin tissue mainly consist of Collagen I, with a small amount of Collagen III (Figure 9A). Collagen III was significantly increased in the untreated scar tissue compared to normal skin tissue (Figure 9B). Collagen I and Collagen III were significantly reduced in the scar tissue treated with triamcinolone injection or the MN patch loaded with rhIFNα-1b compared to the untreated scar tissue (Figure 9C,D). The treated scar tissue was dominated by Collagen I fibers with a small amount of fine filamentous Collagen III fibers. The arrangement of Collagen I fibers and Collagen III fibers became orderly. The MN patch loaded with rhIFNα-1b and the triamcinolone injection both significantly inhibited Collagen I and Collagen III deposition (Figure 10A,B).

### 3.9. Gene Expression

Collagen I, Collagen III, TGF-β1, and α-SMA have been widely recognized as being associated with wound healing and hyperplastic scar formation. Collagen I and Collagen III are the two major collagens present in human skin and are both increased during scar formation. Myofibroblasts are considered to be the main effector cells in scar formation and have a role in promoting collagen synthesis [46]. α-SMA is expressed by fibroblasts and is associated with wound contraction and fibrosis. TGF-β1 promotes fibroblast differentiation and accelerates the conversion of fibroblasts to myofibroblasts [23]. TGF-β1 regulates various fibrosis-related proteins, including Collagen I and Collagen III, during transcription [23]. The relative expressions of Collagen I, Collagen III, TGF-β1, and α-SMA were significantly downregulated in scar tissues treated with the reference preparation and the MN patch loaded with rhIFNα-1b compared to untreated scar tissues (Figure 11A–D). There was no significant difference in the relative expression of indicator genes in scar tissue treated with the reference preparation and the MN patch loaded with rhIFNα-1b.

## 4. Conclusions

In this study, we developed a double-layered soluble polymer MN patch loaded with rhIFNα-1b. A comprehensive study from the design of the MN patch to animal studies was provided. During MN fabrication, the solution containing rhIFNα-1b was concentrated in the MN tips under negative pressure. The MN patch loaded with rhIFNα-1b had sufficient mechanical strength to puncture the skin and to deliver rhIFNα-1b to the epidermis and dermis. The rhIFNα-1b had a significant inhibitory effect on the abnormal proliferation of fibroblasts and excessive deposition of collagen fibers in the scar tissue. The color and thickness of the scar tissue treated using the MN patches loaded with rhIFNα-1b were effectively reduced. The relative expressions of Collagen I, Collagen III, TGF-β1, and α-SMA were significantly downregulated in scar tissues. The MN patch loaded with rhIFNα-1b is a safe and effective way to deliver rhIFNα-1b transdermally.

## Figures and Tables

**Figure 1 polymers-15-02621-f001:**
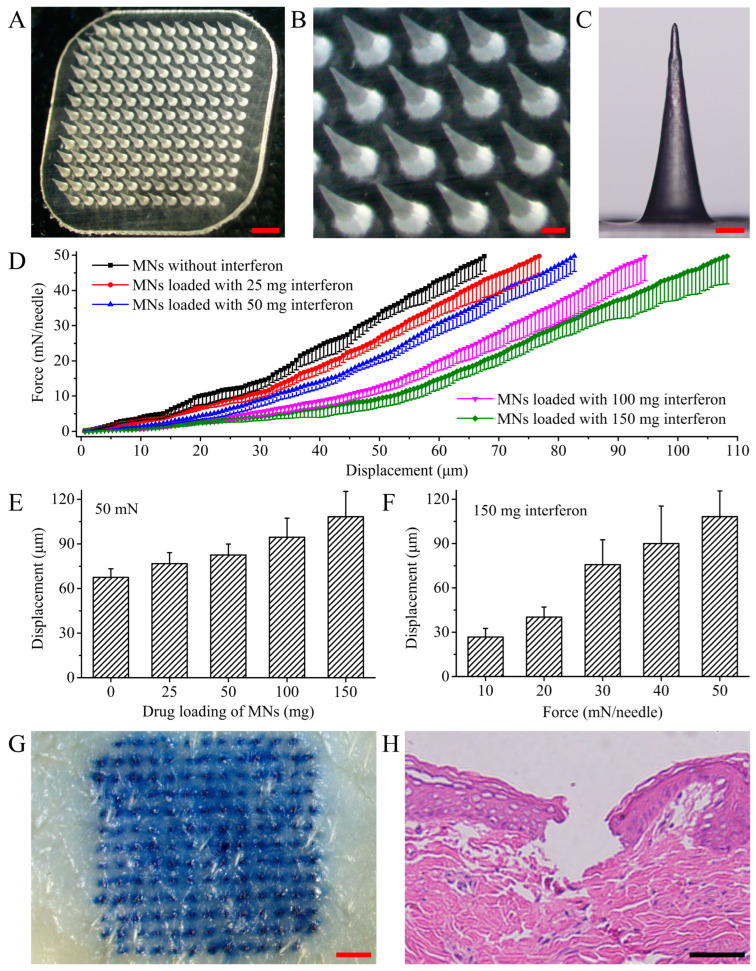
Shape and mechanical strength of the MNs. (**A**,**B**) Optical microscope images of the MN patch. (**A**) Scale bar = 1 mm. (**B**) Scale bar = 200 μm. (**C**) Optical microscope images of MN. (**D**) Force-displacement curves of MNs loaded with different levels of rhIFNα-1b. (**E**) Displacement of MNs loaded with different levels of rhIFNα-1b under 50 mN/needle force. (**F**) Displacement of MNs loaded with 150 mg rhIFNα-1b under different forces. (**G**) Optical microscope images of isolated porcine ear skin punctured by an MN patch loaded with 150 mg rhIFNα-1b (scale bar = 1 mm). (**H**) Optical microscope image of a paraffin section of the skin stained with HE dye (scale bar = 100 µm).

**Figure 2 polymers-15-02621-f002:**
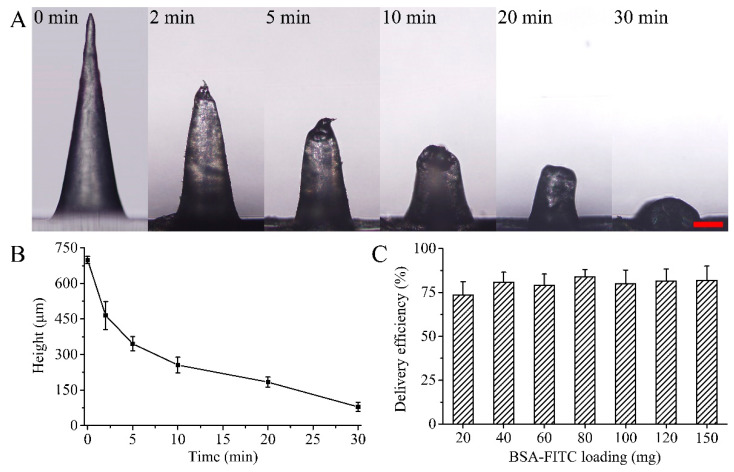
Dissolution rate and delivery efficiency of the MNs in the abdominal skin of ICR mice. (**A**) Optical microscope images of the MNs implanted in the abdominal skin of ICR mice and maintained for 0, 2, 5, 10, 20, and 30 min. (**B**) Variation in MN height with the time of MN patch application on the abdominal skin of male ICR mice. (**C**) Delivery efficiency of MN patches loaded with 20–150 mg BSA-FITC.

**Figure 3 polymers-15-02621-f003:**
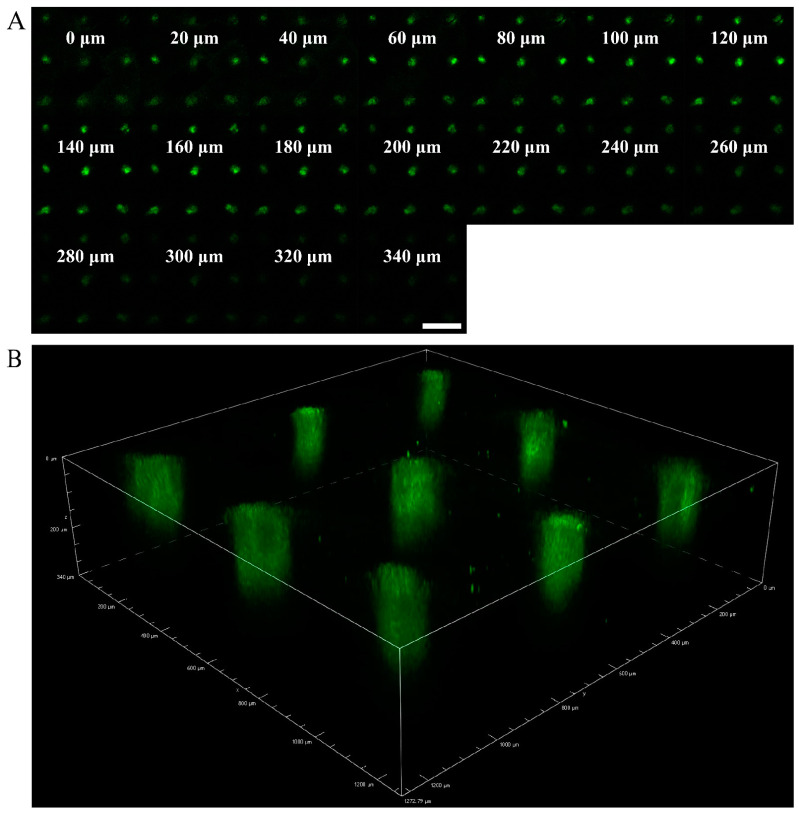
Depth of skin punctured by MNs loaded with BSA-FITC. (**A**) Fluorescence imaging of porcine ear skin punctured by the MNs. (**B**) A 3D reconstruction image of the MN tips implantation site.

**Figure 4 polymers-15-02621-f004:**
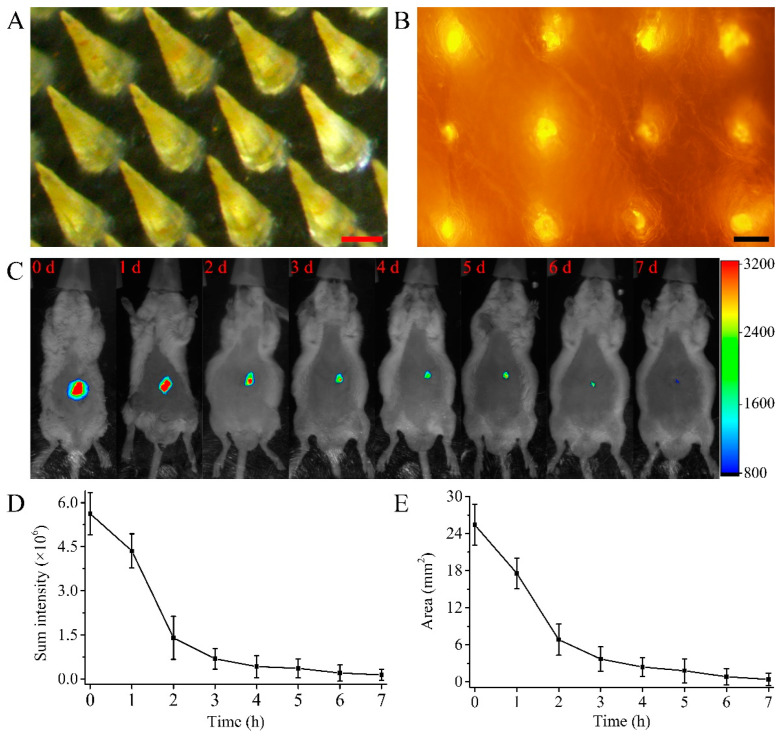
In vivo release of MNs loaded with BSA-FITC. (**A**) Optical microscope image of MNs loaded with BSA-FITC (scale bar = 200 μm). (**B**) Fluorescence microscope image of the abdominal skin of ICR mice punctured by MNs loaded with BSA-FITC (scale bar = 100 μm). (**C**) In vivo imaging of the MNs implanted in the abdominal skin of ICR mice (n = 6). (**D**) Curves of the sum fluorescence intensity versus time. (**E**) Curves of fluorescent area versus time.

**Figure 5 polymers-15-02621-f005:**
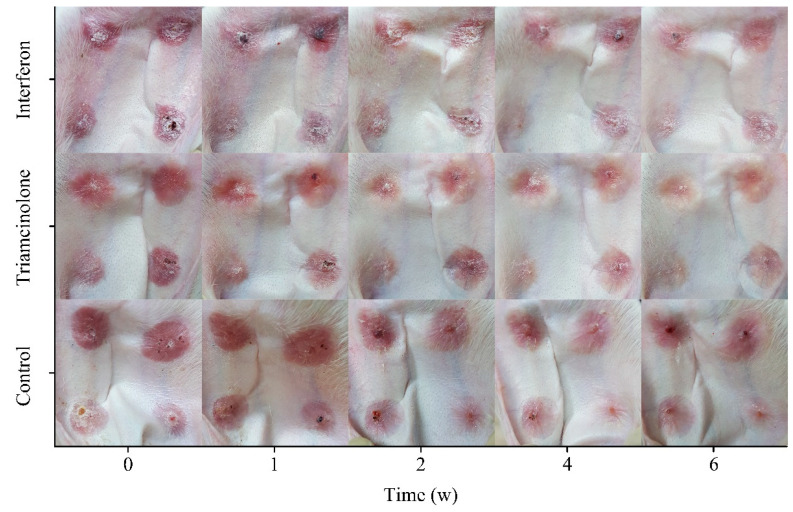
Optical microscope images of rabbit ear scar tissue within 6 weeks.

**Figure 6 polymers-15-02621-f006:**
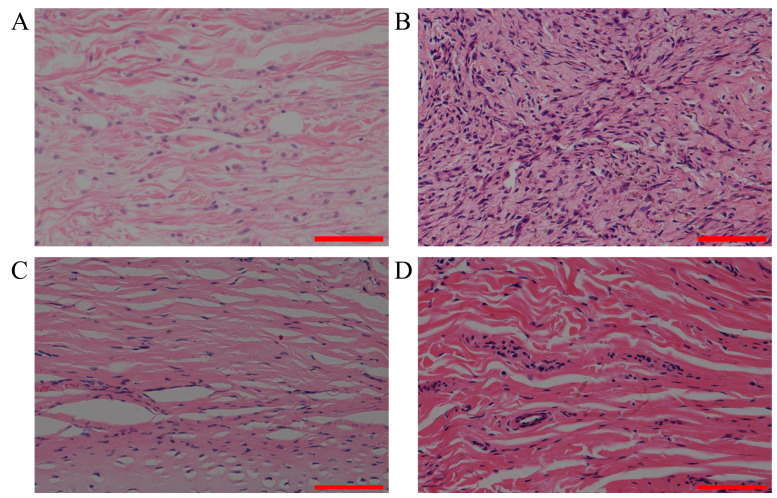
Optical microscope images of paraffin sections of the rabbit ear scar tissue stained with HE dye after 6 weeks of treatment. (**A**) Normal skin tissue (scale bar: 100 μm). (**B**) Untreated scar tissue (scale bar: 100 μm). (**C**) Scar tissue treated with triamcinolone injection (scale bar: 100 μm). (**D**) Scar tissue treated with MN patch loaded with rhIFNα-1b (scale bar: 100 μm).

**Figure 7 polymers-15-02621-f007:**
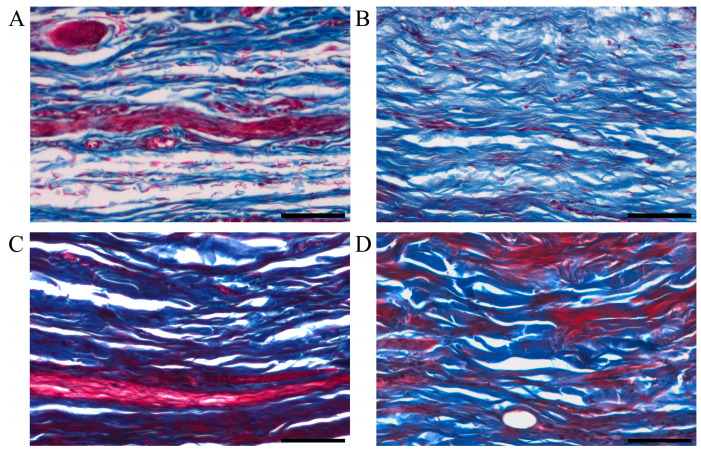
Optical microscope images of paraffin sections of the rabbit ear scar tissue stained with Masson’s trichrome after 6 weeks of treatment. (**A**) Normal skin tissue (scale bar: 100 μm). (**B**) Untreated scar tissue (scale bar: 100 μm). (**C**) Scar tissue treated with triamcinolone injection (scale bar: 100 μm). (**D**) Scar tissue treated with MN patch loaded with rhIFNα-1b (scale bar: 100 μm).

**Figure 8 polymers-15-02621-f008:**
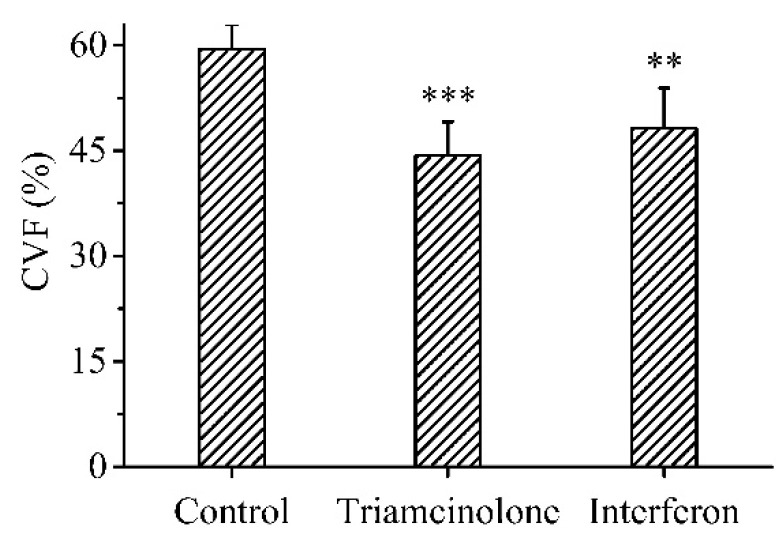
Collagen volume fraction in rabbit ear scar tissue (n = 6). The scar tissue of the control group was not treated in any way. *p* < 0.05 was considered statistically significant, where *p* < 0.01 (**), and *p* < 0.001 (***).

**Figure 9 polymers-15-02621-f009:**
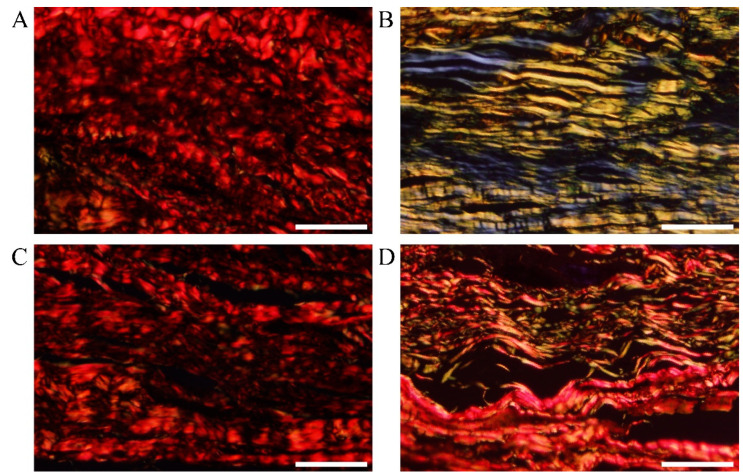
Polarizing microscope images of paraffin sections of the rabbit ear scar tissue stained with Sirius Red after 6 weeks of treatment. (**A**) Normal skin tissue (scale bar: 100 μm). (**B**) Untreated scar tissue (scale bar: 100 μm). (**C**) Scar tissue treated with triamcinolone injection (scale bar: 100 μm). (**D**) Scar tissue treated with MN patch loaded with rhIFNα-1b (scale bar: 100 μm).

**Figure 10 polymers-15-02621-f010:**
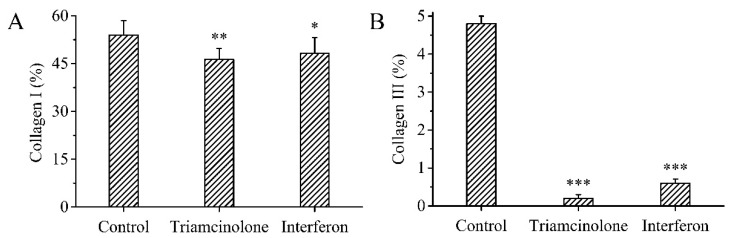
Semi-quantitative analysis of Collagen I (**A**) and Collagen III (**B**) in rabbit ear scar tissue (n = 6). *p* < 0.05 was considered statistically significant, where *p* < 0.05 (*), *p* < 0.01 (**), and *p* < 0.001 (***).

**Figure 11 polymers-15-02621-f011:**
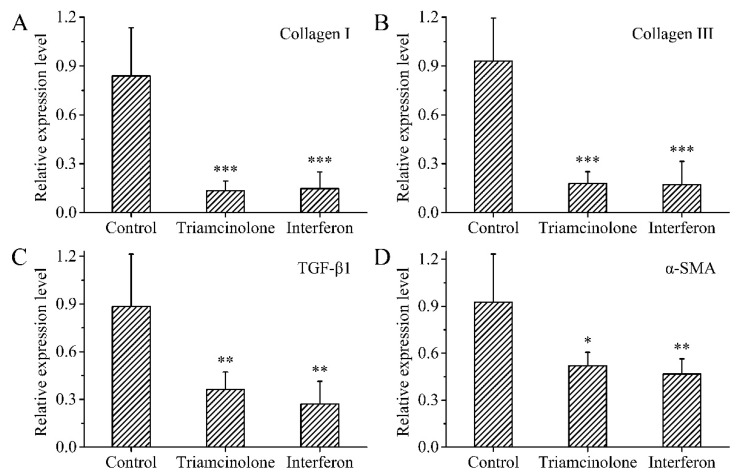
Relative expression levels of genes associated with scarring. (**A**) Collagen I, (**B**) Collagen III, (**C**) TGF-β1, (**D**) α-SMA (n = 6). *p* < 0.05 was considered statistically significant, where *p* < 0.05 (*), *p* < 0.01 (**), and *p* < 0.001 (***).

**Table 1 polymers-15-02621-t001:** Primer design for genes associated with scarring.

Encoding Gene Name	Forward Primer (3′–5′)	Reverse Primer (3′–5′)	Fragment Length (bp)
GAPDH	CTGGTCATCAACGGGAAGGC	CTCCATGGTGGTGAAGACGC	120
α-SMA	GCACTGTCAGGAATCCCGTG	CGGAGCCATTGTCACACACA	83
TGF-β1	GCGGCAGCTGTACATTGACT	TTGTACAGGGCCAGGACCTT	147
Collagen I	GGCCGAACTGGAGAAACAGG	AGCAGTACCAGCCTCTCCAG	81
Collagen III	CCGAACCGTGCCAAATATGC	AACAGTGCGGGGAGTAGTTG	158

## Data Availability

The data in the text are available from the corresponding authors.

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
