# Peer review of "Soluble Polymer Microneedles Loaded with Interferon Alpha 1b for Treatment of Hyperplastic Scar"

_polymers, 2023, doi:10.3390/polym15122621_

Round 1

Reviewer 1 Report

The present manuscript presents a study regarding the obtaining and characterization of microneedles loaded with interferon alpha 1b for treatment of hyperplastic scar. 

The introduction is well structured and the advantages of using microneedles for transdermal release are pointed out, as well as the necessity of using them for the release of interferon alpha 1b for the treatment of hyperplastic scar.  However, the manuscript requires some improvements in terms of obtaining and characterization methods.

The dimensions of the mold and the number or shape of pinholes must be specified in section 2.2. MN fabrication process, not in section 3.1. MN shape and mechanical strength. In Section 3.1. MN shape and mechanical strength data on the shape and size of the obtained MNs must be presented.

In section 2.4 - In vitro release, it is presented at the beginning that the tests will be done on mice, which is not in vitro testing, but in vivo. There is also a phrase about the implantation of MNs into the porcine ear skin, which is actually the release evaluation method? 

To evaluate the release of an active principle, a microscopic view is not enough. The correct evaluation of a drug release is done by quantitative methods. The authors are advised to evaluate the in vitro release of interferon from MNs, using a quantitative method.

In addition, the fluorescent principle chosen to replace interferon has a higher molecular weight, which can lead to erroneous results. An explanation regarding the choice of this fluorescent principle is necessary.

Reviewer 2 Report

Polymers-2406644

Article:  Soluble polymer microneedles loaded with interferon alpha 1b for treatment of hyperplastic scar

Overview

This research work describes an interesting system such as microneedles (composed of a soluble CMC polymer) as an agent to apply interferon alpha 1b in the treatment of hyperplastic scars. This is quite interesting work. I really liked the approach. Next, I will present the view that someone who works with in vitro and in vivo assays. I have just a few suggestions to contribute to the authors. Next, I will make my remarks.

Abstract

The text is clear and objective. It has the essential data present in the paper. In my opinion, it does not need to be modified.

Introduction

The Introduction is well-written and well-contextualized. The main points of this article are objectively presented.

A single shape suggestion. Although the abbreviation MN is in the abstract, in the main text it appears for the first time on page 2 (line 56). I suggest writing in full and putting the abbreviation next. It looks more elegant in the main text.

Materials and methods

The Material and Methods, in my opinion, is very well written. I confess that I am not familiar with all the methodologies used, but I work with most of them. The text allows the reproduction of the essays, so I understand that it fully meets what it proposes.

The methodology adopted also seems to me adequate to answer the questions posed.

A single comment is whether the authors do not consider that there should be some other hypothesis test besides analysis of variance (ANOVA). This can enhance the data and interpretation.

Results and discussion

The results and discussion are very well presented. The data are clear and the cited references are recent. Also, the discussion seems compatible with the results obtained.

My only suggestion is a reflection on identifying the types of collagen by the birefringence associated with picrosirius. Picrosirius is a good indicator for collagen. It allows identifying, and even quantifying, collagen in a simple and very reliable way. But separating collagen types by birefringence has been questioned. Personally, I understand that this is a very reasonable doubt. This does not modify the content of the results and brings about the need to change the figures, but it implies rewriting some points. Authors are invited to consult:

- Vidal et al. (1982) Histochem J.;14(6):857-78. doi: 10.1007/BF01005229

- Dayan et al (1989) Histochemistry; 93(1):27-9. doi: 10.1007/BF00266843

- Dapson et al (2011) Biotech Histochem.; 86(3):133-9. doi: 10.3109/10520295.2011.570277

The authors will see that changes in birefringence may also be associated with the thickness and level of collagen aggregation. This new view may even be more consistent with the interpretation that collagen III is not increased (which can be seen in gene expression assays). I leave this reflection to the authors.

From a shape point of view, figures 6D and 7D could have their brightness adjusted to be more consistent with the others presented. But this does not interfere with the quality of the results.

Conclusion

I understand that the conclusions presented in the paper are in line with the data presented.

References

The references are quite recent. Congratulations to the authors.

Round 2

Reviewer 1 Report

The authors answered only part of the indicated requirements. Section 2.4 should be revised and the methods used, both in vivo and ex vivo, explained in more detail. It should be pointed out in the manuscript the necessity of using ex vivo tests on the pig's ear and the section can be divided into: ex vivo, in vivo dissolution tests, and in vivo release tests.

Author Response

Point 1: The authors answered only part of the indicated requirements. Section 2.4 should be revised and the methods used, both in vivo and ex vivo, explained in more detail. It should be pointed out in the manuscript the necessity of using ex vivo tests on the pig's ear and the section can be divided into: ex vivo, in vivo dissolution tests, and in vivo release tests.

Response 1: We greatly appreciate your careful comments on our work. In the revised manuscript, section 2.4 has been divided into in vivo dissolution test and in vivo release test. Among them, the in vivo dissolution test was performed on the abdominal skin of male ICR mice. "The MNs were implanted into the porcine ear skin" has been revised to "The MNs were implanted into the abdominal skin of male ICR mice". Previously, our team evaluated the dissolution rate of the microneedles by implanting them into isolated porcine ear skin. However, considering the difference in the dissolution rate of the microneedles in ex vivo and in vivo skin tissue. Finally, we performed in vivo dissolution test using the abdominal skin of male ICR mice. Thank you again for your comments. We apologize for the clerical error here.
